

# Allopatric speciation is more prevalent than parapatric ecological divergence in a recent high-Andean diversification (*Linochilus:* Asteraceae)

Oscar M. Vargas[1,2], Santiago Madriñán[3,4] and Beryl Simpson[2]

[1] Department of Biological Sciences, California State Polytechnic University, Humboldt, Arcata, CA, United States
[2] Department of Integrative Biology and Billie Turner Plant Resources Center, The University of Texas at Austin, Austin, TX, USA
[3] Department of Biological Sciences, University of the Andes, Bogotá, DC, Colombia
[4] Jardín Botánico de Cartagena, Turbaco, Bolívar, Colombia

Corresponding author
Oscar M. Vargas,
oscarvargash@gmail.com

## ABSTRACT

Elucidating how species accumulate in diversity hotspots is an ongoing debate in evolutionary biology. The páramo, in the Northern Andes, has remarkably high indices of plant diversity, endemicity, and diversification rates. A hypothesis for explaining such indices is that allopatric speciation is high in the páramo given its island-like distribution. An alternative hypothesis is that the altitudinal gradient of the Andean topography provides a variety of niches that drive vertical parapatric ecological speciation. A formal test for evaluating the relative roles of allopatric and parapatric ecological speciation is lacking. The main aim of our study is to test which kind of speciation is more common in an endemic páramo genus. We developed a framework incorporating phylogenetics, species' distributions, and a morpho-ecological trait (leaf area) to compare sister species and infer whether allopatric or parapatric ecological divergence caused their speciation. We applied our framework to the species-rich genus *Linochilus* (63 spp.) and found that the majority of recent speciation events in it (12 events, 80%) have been driven by allopatric speciation, while a smaller fraction (one event, 6.7%) is attributed to parapatric ecological speciation; two pairs of sister species produced inconclusive results (13.3%). We conclude that páramo autochthonous (*in-situ*) diversification has been primarily driven by allopatric speciation.

## INTRODUCTION

Alexander von Humboldt and Aimé Bonpland's *Tableau Physique des Andes et Pays Voisins* (1805) illustrated how plant species are assembled from lowlands to high elevations in the tropical Andes. This representation visualized for the first time how diversity changes locally with elevation, setting a seminal starting point for biodiversity studies. Today, 218 years after the first publication of the *Essay on the Geography of Plants* (*von*

*Humboldt & Bonpland, 1805*), scientists have mapped species richness around the globe and have identified numerous biodiversity hotspots. Understanding how species accumulation occurs in such hotspots is pivotal for the fields of evolutionary biology and biogeography.

Because biodiversity hotspots often coincide with areas of topographic complexity (*Barthlott, Lauer & Placke, 1996*; *Myers et al., 2000*; *Mutke & Barthlott, 2005*; *Jenkins, Pimm & Joppa, 2013*; *Rahbek et al., 2019a*, *2019b*), geographical isolation and ecological opportunity are typically cited to explain species richness. In mountain systems, valleys and canyons act as landscape barriers for some organisms (*Janzen, 1967*; *van der Hammen & Cleef, 1986*; *Muños-Ortiz et al., 2015*), promoting allopatric speciation (vicariant or dispersal) *via* geographical isolation, while slopes and ecological gradients can drive parapatric speciation *via* ecological divergence (*Hughes & Atchison, 2015*; *Pyron et al., 2015*). Vicariant speciation occurs when a mother species that is distributed broadly is divided into daughter populations that speciate *via* subsequent independent evolution. Glacial and interglacial cycles are often assumed to have caused such vicariant events (*van der Hammen & Cleef, 1986*; *Carstens & Knowles, 2007*; *Flantua et al., 2019*). Peripatric speciation occurs when a dispersal event from a source lineage to a new previously uncolonized area takes place (founder speciation event). If the newly colonized area is geographically isolated enough, over time, lineages can become reproductively isolated from one another, resulting in progenitor and derivative lineages (*Coyne & Orr, 2004*). Parapatric speciation takes place when a continuous population that is distributed along an ecological gradient (*e.g.*, an altitudinal gradient) is subdivided in two or more subpopulations that locally adapt to different niches (*e.g.*, lower *vs.* higher elevations); with subpopulations becoming independent species by means of ecological differentiation and non-random mating, leading to the formation of reproductive barriers (*Givnish, 1997*; *Schluter, 2000*; *Simpson, 1953*).

Sister species, two species that recently diverged from a common ancestor, provide an opportunity to test for the predictions of different speciation types. Under a scenario of allopatric speciation, recently diverged species should be similar to each other due to phylogenetic conservatism (Fig. 1A). In the context of mountain-tops (sky-islands), sister species that arose because of geographic speciation should occupy similar niches on different islands and have similar ecological characteristics (*Pyron et al., 2015*). Alternatively, under a scenario of parapatric ecological speciation, sister species are expected to occupy different niches within an island, presenting a signature of ecological divergence (Fig. 1D) (*Pyron et al., 2015*).

The páramo is a high-elevation ecosystem found above the timberline (~3,000 m, Fig. 2) primarily in the Northern Andes (Ecuador, Colombia, and Venezuela) (*Cuatrecasas, 1968*; *Luebert & Weigend, 2014*; *Weigend, 2002*). With ca. ~3,400 species of vascular plants (*Luteyn, 1999*), of which 60% to 100% are estimated to be endemic (*Luteyn, 1992*; *Madriñán, Cortés & Richardson, 2013*), and particularly high diversification rates (*Madriñán, Cortés & Richardson, 2013*), the páramo is considered the most species rich ecosystem of the world's tropical montane regions (*Sklenář, Hedberg & Cleef, 2014*).
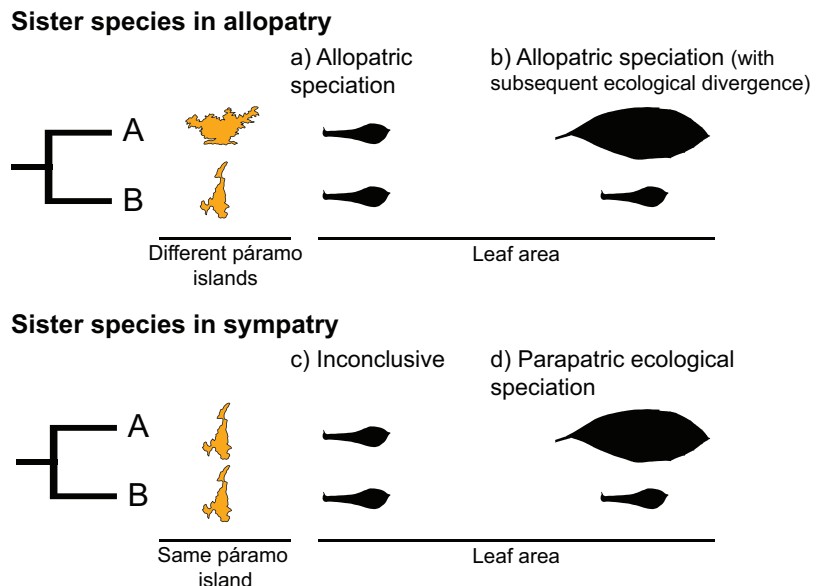

**Sister species in allopatry**

a) Allopatric speciation

b) Allopatric speciation (with subsequent ecological divergence)

A

B

Different páramo islands

Leaf area

**Sister species in sympatry**

c) Inconclusive

d) Parapatric ecological speciation

A

B

Same páramo island

Leaf area

**Figure 1 Hypothetical scenarios for speciation in sister species.** (A) Allopatric speciation event in which geographical isolation resulted in two species living in separate páramo islands (indicated by island shape) and occupying similar niches (as indicated by similar leaf areas). (B) Allopatric speciation event in which geographical isolation resulted in two species living in separate páramo islands occupying different niches as indicated by different leaf areas. (C) A speciation event in which the reason for divergence is inconclusive, sister species inhabit the same island and have similar leaf areas. (D) A parapatric ecological speciation event in which sister species evolved different leaf areas in response to selection to different niches on the same páramo island.

Testing how speciation happens in the páramo would help to elucidate the processes that lead to accumulation of species in high elevation, island-like, biodiversity hotspots.

Unlike other South American high-elevation areas farther south (*i.e.*, Perú and Chile), the páramo is characterized by abundant precipitation throughout the year, and is therefore defined by both high elevation and humidity. It has been estimated that the páramo originated when the high elevations (>3,000 m) of the Northern Andes emerged as a result of rapid uplift 2–4 mya (*van der Hammen & Cleef, 1986*; *Gregory-Wodzicki, 2000*). The fragmented nature of the páramo (Fig. 2) and its altitudinal gradient (~3,000–4,500 m) are hypothesized to have acted as drivers of diversification by promoting both allopatric speciation (vicariant or dispersal) *via* geographical isolation (*van der Hammen & Cleef, 1986*) and parapatric speciation *via* ecological divergence (*Hughes & Atchison, 2015*; *Nevado et al., 2018*). Despite efforts documenting Andean diversification (*e.g.*, *Nürk et al., 2015*; *Uribe-Convers & Tank, 2015*; *Lagomarsino et al., 2016*; *Pérez-Escobar et al., 2017*) a formal test to quantify the relative prevalence of allopatric *vs.* parapatric ecological speciation of taxa in the region is lacking.

To quantify the relative contributions of allopatric and parapatric speciation in the páramo, we used a comparative framework that combines phylogenetic, geographical, and ecological information. We applied this approach to *Linochilus* (Asteraceae: Astereae), a genus restricted to the páramo and the upper boundary of the cloud forest (*Blake, 1928*; *Cuatrecasas, 1969*; *Vargas, 2011*, *2018*; *Vargas, Ortiz & Simpson, 2017*; *Saldivia et al.,*

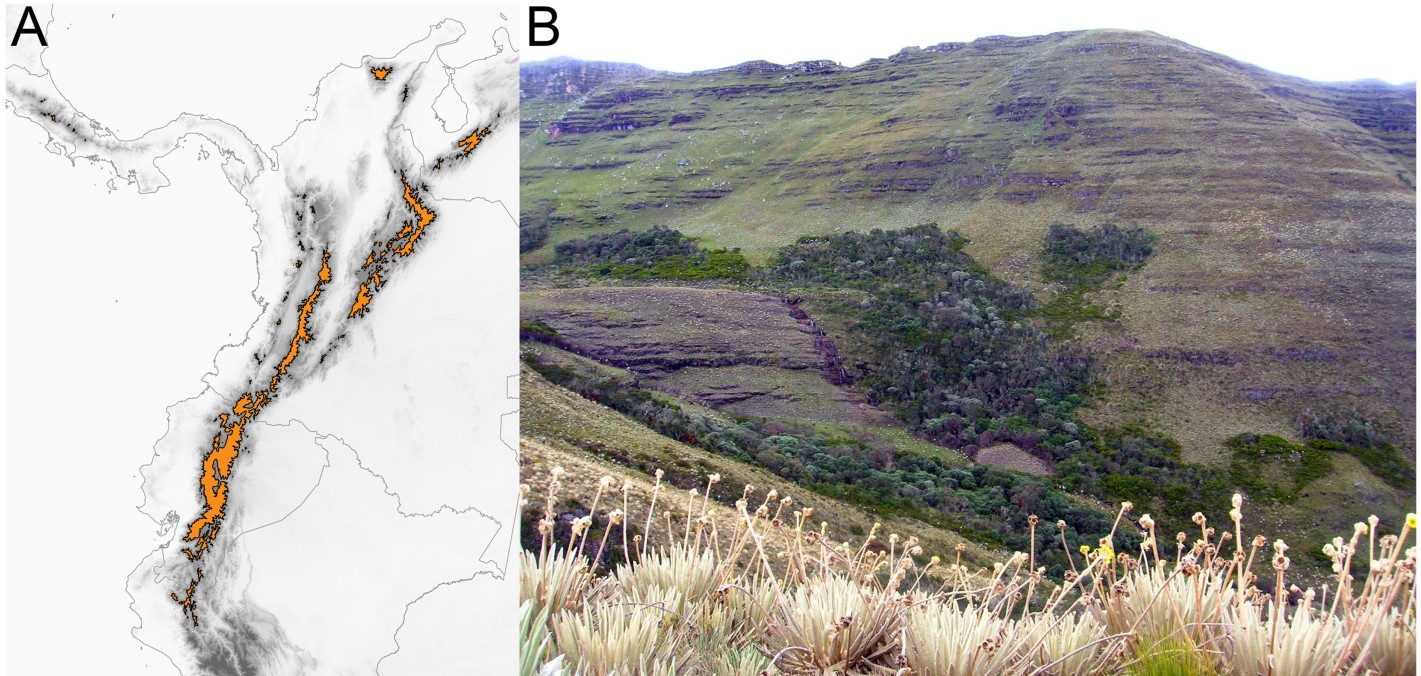

**Figure 2 The páramo ecosystem.** (A) Approximate geographical extent of the páramo ecosystem. (B) Páramo la Rusia, Boyacá, Colombia; notice how high the Andean forest interdigitates with the páramo along the creek.

*2019*). *Lichochilus'* phylogeny was recently inferred using high-throughput sequencing (*Vargas, Ortiz & Simpson, 2017*), and a taxonomic monograph is almost complete for the genus by OMV. *Lichochilus* was recently segregated from *Diplostephium* because *Diplostephium* s.l. is biphyletic. *Linochilus* is a Northern Andean clade sister to a clade that comprises numerous genera including *Baccharis* and *Diplostephium* s.s., the latter with similar morphology and ecology primarily inhabiting high elevations in the Central Andes (*Vargas, Ortiz & Simpson, 2017*; *Vargas, 2018*; *Saldivia et al., 2019*).

The main aim of this study is to ascertain which kind of speciation, allopatric *vs.* parapatric ecological divergence, is more common in this plant genus that inhabits island-like elevational areas and their lower boundaries. We specifically aim to (1) quantify the relative contribution of allopatric and parapatric ecological speciation in the divergence of sister taxa in *Linochilus*, and (2) to pinpoint the geographical origin of *Linochilus* employing a historical biogeographical analysis.

## MATERIALS AND METHODS

### Focal clade

*Linochilus* (Asteraceae: Astereae) inhabits primarily the páramo, although some species dwell in the upper boundary of the cloud forest due to a downslope colonization event (*Vargas & Madriñán, 2012*). We studied this genus because of the comprehensive understanding of the distribution, taxonomy (*Cuatrecasas, 1969*; *Vargas, 2011*; *Vargas, 2018*, Vargas in prep), and phylogeny of its species (*Vargas, Ortiz & Simpson, 2017*). A phylogenetic study lead by the first author (*Vargas, Ortiz & Simpson, 2017*), used genome

skimming and ddRAD sequencing, including 36 (57%) out of the 63 known species. *Linochilus* is distributed in the disjunct mountains of the Talamanca Cordillera (Costa Rica), the Sierra Nevada de Santa Marta (Colombia), and the Northern Andes (Colombia, Venezuela, and Ecuador), exhibiting a variety of woody habits from decumbent subshrubs only 10 cm tall to small trees 6 m tall (*Cuatrecasas, 1969*; *Vargas, 2018*). Growth form and leaf area of *Linochilus* species are associated with the habitat they occupy—shrubs and decumbent subshrubs with microphyllous leaves inhabit the open páramo, while small trees with broad leaves reside at lower elevations in the upper, more humid, edge of the Andean forest (*Vargas & Madriñán, 2012*). Broad-leaf forest species comprise the Denticulata clade (*Cuatrecasas, 1969*; *Vargas, 2018*).

## Sister species comparisons

To measure the relative contribution of allopatric speciation and parapatric ecological divergence to recent speciation events, we compared the geographical distribution (Appendix S1) and the leaf areas (Appendix S2) of putative sister species based on phylogeny and taxonomy (see assumption section below). We used leaf area as a proxy to evaluate ecological divergence between these sister species pairs. Leaf area is a functional character that varies with the eco-physiological pressures of a species' niche, reflecting its adaptation to water availably, irradiance, and elevation (*Givnish, 1987*; *Reich et al., 1999*), therefore providing a proxy for ecological niche. For example, *L. antioquensis* inhabits the upper Andean forest and has an average leaf area of 848.4 $mm^2$. In contrast, *L. phylicoides*, dwells in the physiologically dry open páramo with plenty of access to sunlight, having an average leaf area of 4.0 $mm^2$. When possible, we measured the area of 30 leaves from six different individuals for each species (Appendix S3). All measurements were done in adult (flowering) plants. We scanned the leaves at 600 dpi from herbarium material belonging to ANDES, TEX, and US herbaria. Each leaf was outlined using PHOTOSHOP CS4 (Adobe Systems, San Jose, CA, USA) as a single white and black image. We then used the R package MOMOCS (*Bonhomme et al., 2014*) to calculate the area of each leaf from the images created in the previous step. Because leaf area increases exponentially, we transformed the data logarithmically. We performed a Wilcoxon signed-rank tests of the log-transformed leaf areas between sister species using R (*R Core Team, 2016*). We also used the Wilcoxon signed-rank test to compare the leaf area between the Denticulata and its sister clade, and between Denticulata and *Linochilus*'s most species-rich clade (the clade originating with the most common ancestor of *L. heterophyllus* and *L. phylicoides*). It is important to note that this test considers the distribution of values for leaf areas in each species (instead of just the average) and can be used to compare non-independent samples.

We used and compared two different approaches to score species pairs as allopatric or sympatric. First, using the páramo delineation of *Londoño, Cleef & Madriñán (2014)*, we scored a pair of species as allopatric when they inhabited non-overlapping páramo islands (mountaintops) and/or when they inhabited different slopes of the same mountain (this pattern is possible for non-páramo species that inhabit the upper boundaries of the cloud forest). We scored a pair as sympatric when their distribution overlapped at least one páramo island—in other words, the presence of the two sisters in a single páramo island

were considered sympatric. Second, we calculated range overlap and range asymmetry between sisters by overlaying occurrences in a grid following *Vargas et al. (2020)*. We used two grid sizes, 0.05 and 0.1 decimal degrees, corresponding to 33 and 131 Km$^2$ respectively. With the grid-ranges of sister species, we calculated the sisters' overlap as the area occupied by both sisters divided by the summed area of the smaller ranged sister. An overlap of 0 indicated full allopatry while an overlap of 1 indicated that the smaller-ranged sister was found solely within the range of its sister (*Barraclough & Vogler, 2000*; *Fitzpatrick & Turelli, 2006*). Range asymmetry was calculated as the area occupied by the larger-ranged sister divided by the area of its sister (*Fitzpatrick & Turelli, 2006*). The aforementioned calculations were made on R using the "raster" package v. 3.5-21 (*Hijmans, 2022*). Distributional data were extracted from curated COL and US herbaria specimens.

We interpreted the results as follows (Fig. 1):

- If a sister-species pair is allopatric and there is no significant difference between their leaf areas, we interpreted this scenario as an event of allopatric speciation driven by geographical isolation, with the reasoning that leaf area had remained similar because either relatively little time had passed since divergence and/or because of niche conservatism (Fig. 1A; *Wiens, 2004*; *Pyron et al., 2015*).

- If a sister-species pair is allopatric and their leaf areas are significantly different, we interpreted this scenario as an event of allopatric speciation following geographical isolation (Fig. 1B) in which there was subsequent ecological divergence driven by local adaption (*Rundell & Price, 2009*; *Pyron et al., 2015*).

- If a sister-species pair has overlapping geographical distributions and there is no significant difference between their leaf areas, we interpreted this scenario as inconclusive (Fig. 1C). This pattern could be the result of various processes, *i.e.*, sympatric speciation, allopatric speciation with no ecological divergence followed by secondary contact (*Rundell & Price, 2009*; *Hopkins, 2013*), or parapatric speciation driven by ecological divergence in a trait other than leaf area (*e.g.*, *Snaydon & Davies, 1976*; *Silvertown et al., 2005*).

- If a sister-species pair is scored as sympatric and their leaf areas are different, we interpreted this scenario as an event of parapatric speciation with ecological divergence (*Rundle & Nosil, 2005*; *Rundell & Price, 2009*) (Fig. 1D).

We acknowledge that the role of parapatric ecological divergence may be underestimated in this study because we only measured leaf area as an ecological proxy. Ecological traits independent from leaf area can confer the ability to colonize different páramo niches (*e.g.*, *Cortés et al., 2018*), such as underground eco-physiological adaptations to soils with different water saturation, or physiological adaptations at the anatomical and cellular level. Alternative physio-ecological variables in sister species comparisons could shed light on other types of ecological divergence (*e.g.*, adaptations to different moisture in soils, microclimatic preferences) but elevation and difficulties of access to páramos make *in situ* studies difficult.

Our framework assumes that:

1) Leaf area represents a good proxy for the organism's niche, which is likely true in our focal genus, *Linochilus*. Many other plant groups show leaf sizes that are associated with their habitat *i.e.*, broad and large leaves in species that dwell in the upper limit of the Andean forest, medium to small leaves in taxa inhabiting the páramo, and microphyllous leaves in lineages that inhabit the uppermost boundary of the páramo (also known as superpáramo) (*Cuatrecasas, 1969*; *Vargas & Madriñán, 2012*).

2) The phylogeny employed represents the best species tree for the species sampled. The best hypothesis to date for the species tree in *Linochilus* (*Vargas, Ortiz & Simpson, 2017*, based on double digest restriction associated DNA data) contains a subset of documented *Linochilus* species (36 of 63). To account for the potential effects of missing species in our phylogeny, we searched the taxonomic literature for the most morphologically similar species to those not sampled in the phylogeny, identifying alternative sets of species pairs (Appendix S4). Specifically, we searched for descriptions of species where authors indicate the most similar taxon based on key traits. Diagnostic characters in *Linochilus* include, but are not limited to, habit, the size and shape of the leaf, the number of capitula per inflorescence, and measurements in the corolla of both disk and ray flowers. Morphological (*Cuatrecasas, 1969*; *Vargas & Madriñán, 2006*) and phylogenetic studies (*Vargas, Ortiz & Simpson, 2017*) in *Linochilus* largely agree on the grouping of species, for example the *Denticulata* clade groups the *Linochilus* series *Denticulata* and *Huertasina* (*Cuatrecasas, 1969*). Pairs *L. obtusus–L. venezuelensis*, *L. rhododendroides–L. schultzii*, and *L. oblongifolius–L mutiscuanum* are supported as putative sister by both taxonomy and phylogenetics (*Cuatrecasas, 1943*, *1969*; *Vargas, Ortiz & Simpson, 2017*). The set of alternative pairs based on morphology replaced some sisters from the phylogeny-based pair analysis, maintaining a similar number of pairs and making the results comparable. For example, if species A and B were found to be sister in the phylogeny, and C was not sampled but proposed in the taxonomic literature to be sister to A, the pair A C replaced A B in our substituted analysis. We provide two sets of results: (a) considering only the pairs derived from the phylogeny and (b) substituting non-sampled species to their most likely sister species present on the phylogeny (we provide a discussion explaining why we believe this second set of results is more reliable).

3) The páramo developed only in the last 2–4 mya (*van der Hammen & Cleef, 1986*; *Gregory-Wodzicki, 2000*) and it is therefore reasonable to assume that its carrying capacity for the number of species has not been reached. Extremely high diversification rates support the previous statement (*Madriñán, Cortés & Richardson, 2013*).

4) The modern species ranges are representative of past ranges, a reasonable assumption given the recent divergence of sister taxa pairs and slow growth rate expected for these high elevation woody plants.

## Evaluating niche conservatism

We used leaf area as a proxy to infer the ecological niche of *Linochilus* species. To test for phylogenetic signal in the evolution of the leaf area, we averaged log-transformed leaf data and calculated *Pagel's (1999)* lambda ($\lambda$) using the R package PHYTOOLS (*Revell, 2012*).
To calculate λ, the observed phylogeny was compared to modified trees in which the internal branches are compressed to various degrees. When λ = 0 the trait data follow a model in which internal branches are completely collapsed (star phylogeny), meaning that the trait evolves independently from the phylogeny. When λ = 1 the trait data follow a model in which the internal branches are not modified, meaning that the evolution of the trait is depenent on the phylogeny (*Harmon, 2018*; but see *Revell, Harmon & Collar, 2008*).

## Biogeographic analysis

To elucidate the biogeographic history of *Linochilus* species, we performed a historical biogeographic analysis. We defined the biogeographic areas based on the páramo bioregions defined by *Londoño, Cleef & Madriñán (2014)* that employed a floristic parsimony analysis of endemicity on 30 localities of the Colombian páramos. We added three bioregions to complete the distribution of *Linochilus* (and the páramo) following (*Vargas, 2016*). Contours of páramo areas were edited with QGIS 2.8Wien (*QGIS Development Team, 2005*). The areas considered, following *Vargas (2016)*, are (* = areas not included by *Londoño, Cleef & Madriñán (2014)*):

- *Northern Páramos (N).* Páramos in the "Sierra Nevada de Santa Marta" and the "Serranía del Perijá".
- *Talamanca* (T).* Páramos in the Talamanca Cordillera, Central America.
- *Mérida* (T).* Páramos in the Mérida Cordillera, Venezuela.
- *Eastern Cordillera (E).* Páramos in the Colombian Eastern Cordillera.
- *Antioquia (A).* Cluster of páramos comprising the areas in the Colombian Western and Central Cordilleras in the department of Antioquia.
- *Western Cordillera (W).* Páramos in the Colombian Western Cordillera with the exception of those located in the department of Antioquia.
- *Central Cordillera (C).* Páramos in the Colombian Central Cordillera with the exception of those located in the department of Antioquia.
- *Southern Páramos (S).* Páramos in the Colombian Massif and the Ecuadorian Andes.

We pruned the chronogram of *Vargas, Ortiz & Simpson (2017)* to include only *Linochilus* species and used BioGeoBEARS (*Matzke, 2013*) to infer the biogeographic history of the genus. BioGeoBEARS use Maximum Likelihood to evaluate the fit of different models: DIVALIKE (parameters: dispersal and vicariance; *Ronquist, 1997*), DEC (parameters: dispersal, extinction, and narrowly sympatric, budding, and vicariance cladogenesis; *Ree & Smith, 2008*), and BAYAREALIKE (parameters: dispersal, extinction, and narrowly sympatric and widely sympatric cladogenesis; *Landis et al., 2013*). BioGeoBEARS also evaluates the addition of the J parameter (*Matzke, 2014*) to each one of the models to account for founder-event speciation (DEC+J, DIVALIKE+J, BAYAREALIKE+J, see *Ree & Sanmartín (2018)* and *Matzke (2022)* for a discussion about the use of the +J parameter). Model selection was based on the AIC score by different models. We opted not to use a constrained model (*e.g.*, limiting the presence areas to time windows based on their inferred history) because the paleoelevations of the Andes are still

**Table 1 Main results from sister species comparisons.** Relative signal of geographical isolation and parapatric ecological divergence in *Linochilus* based on sister species comparisons.

| Sister spp. pairs based on: | Geographical isolation | Parapatric ecological divergence | Inconclusive |
|---|---|---|---|
| Phylogeny of *Vargas, Ortiz & Simpson (2017)* | 9 (64.3%) | 3 (21.4%) | 2 (14.3%) |
| Phylogeny substituted with missing sampling | 12 (80%) | 1 (6.7%) | 2 (13.3%) |

**Table 2 Sister species comparisons based on the phylogeny of *Vargas, Ortiz & Simpson (2017)* substituted with non-sampled species (underlined).** Allopatry and sympatry is determined based on overlapping occurrences on paramo islands as defined by *Londoño, Cleef & Madriñán (2014)*. Notice that there are two hypothesized sisters for *L. floribundus* in the taxonomic literature, both are allopatric relative to *L. floribundus*.

| Sister 1 | Sister 2 | Wilcoxon test leaf | Distribution | Inferred divergence | Age (mya) |
|---|---|---|---|---|---|
| *L. phylicoides* | *L. lacunosus* | 0.4722 | Allopatric | Geog. isolation | 0.16 (0.01–0.58) |
| *L. obtusus* | *L. venezuelensis* | 1.0000 | Sympatric | Inconclusive | 0.22 (0.01–0.81) |
| *L. violaceus* | *L. cinerascens* | 0.2897 | Allopatric | Geog. isolation | 0.28 (0.01–0.86) |
| *L. rosmarinifolius* | *L. cyparissias* | – | Allopatric | Geog. isolation | – |
| *L. floribundus* | *L. farallonensis/perijaensis* | – | Allopatric | Geog. isolation | – |
| *L. rhomboidalis* | *L. apiculatus* | 2.50E−09* | Allopatric | Geog. isolation | 0.59 (0.01–1.54) |
| *L. alveolatus* | *L. costaricensis* | 1.0000 | Allopatric | Geog. isolation | 0.56 (0.01–1.46) |
| *L. rhododendroides* | *L. schultzii* | 1.0000 | Allopatric | Geog. isolation | 0.92 (0.02–2.14) |
| *L. tenuifolius* | *L. ellipticus* | – | Allopatric | Geog. isolation | – |
| *L. oblongifolius* | *L. mutiscuanus* | 0.4140 | Allopatric | Geog. isolation | 0.20 (0.01–0.76) |
| *L. sp. nov. ANT* | *L. antioquensis* | 0.1663 | Allopatric | Geog. isolation | 0.52 (0.01–1.47) |
| *L. huertasii* | *L. julianii* | – | Allopatric | Geog. isolation | 1.41 (0.05–3.18) |
| *L. eriophorus* | *L. chrysotrichus* | – | Allopatric | Geog. isolation | – |
| *L. colombianus* | *L. glutinosus* | 1.20E−11* | Sympatric | Ecological | 2.97 (0.11–6.17) |
| *L. romeroi* | *L. saxatilis* | – | Sympatric | Inconclusive | – |

**Note:**
*Statistically significant.

debated (*Luebert & Weigend, 2014*). We assigned areas to tips based on the same occurrence dataset used for the sister species comparisons (Appendix S1).

## RESULTS

We evaluated possible speciation scenarios for *Linochilus* sister species inferred from a molecular phylogeny (14 pairs) and a set of sisters considering non-sampled taxa in the phylogeny (15 pairs). We employed different methods to infer range overlap on our sister species comparisons to cross-check robustness. Independently from the set of pairs used and the method to infer range overlap, our results unequivocally reveal a strong signal of geographic isolation with little ecological divergence (Tables 1 and 2, Fig. 3, Appendices S5–S7). When only species included in the *Vargas, Ortiz & Simpson (2017)* phylogeny are considered (row 1, Table 1; Appendices S5–S7), nine out of 14 sister species pairs (64.3%) are allopatric, with only one of these pairs presenting different leaf areas, suggesting divergence by geographical isolation with little ecological divergence after speciation. A

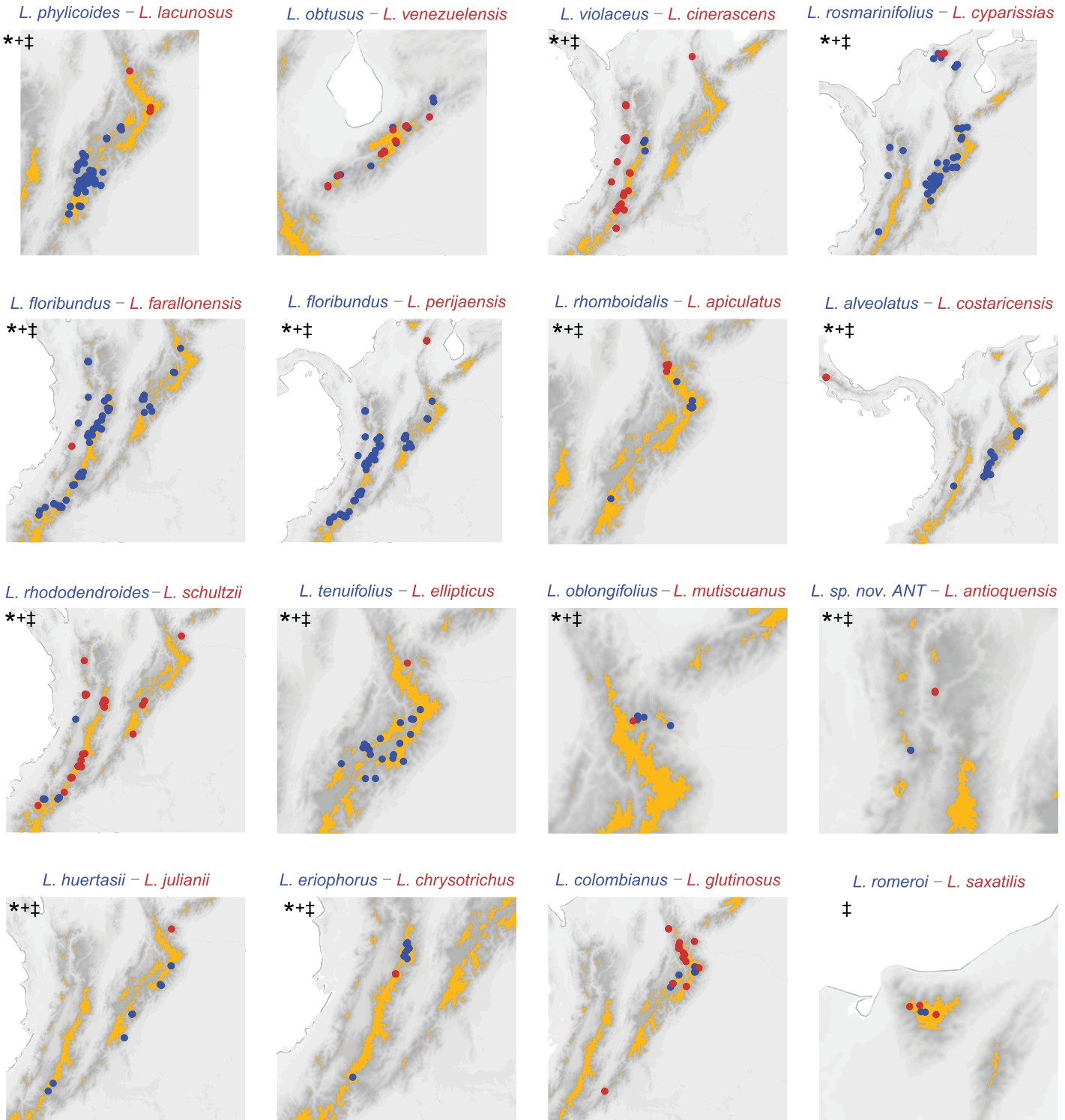

**Figure 3 Distribution of sister species based on the phylogeny (*Vargas, Ortiz & Simpson, 2017*) substituted with missing sampling.** Darker pixels indicate higher elevations, orange polygons delineate páramo regions. *: allopatric based on páramo islands. +: allopatric based on 0.1 decimal degree grid analysis. ‡: allopatric based on 0.05 decimal degree grid analysis. Pairs *L. rosmarinifolius–L. cyparissias* and *L. oblongifolius–L. mutiscuanus*, all inhabiting upper boundary of the cloud forest, are codified as allopatric in the island framework as they inhabit different slopes in the mountains they co-occur.

**Table 3 Range overlap calculated using 0.1 and 0.05 decimal degrees grids.** *Linochilus* sister species comparisons based on the phylogeny of *Vargas, Ortiz & Simpson (2017)* substituted with non-sampled species (underlined). *L. floribundus* is found twice in the table because there are two hypothesized sisters for it in the taxonomic literature.

| Sister 1 | Sister 2 | Range overlap | | Range asymmetry | |
|---|---|---|---|---|---|
| | | 0.1 | 0.05 | 0.1 | 0.05 |
| *L. phylicoides* | *L. lacunosus* | 0 | 0 | 10.7 | 16.4 |
| *L. obtusus* | *L. venezuelensis* | 0.4 | 0.2 | 1.0 | 1.1 |
| *L. violaceus* | *L. cinerascens* | 0 | 0 | 5.7 | 6.3 |
| *L. rosmarinifolius* | *L. cyparissias* | 0 | 0 | 21.8 | 25.3 |
| *L. floribundus* | *L. farallonensis* | 0 | 0 | 42.0 | 46.0 |
| *L. floribundus* | *L. perijaensis* | 0 | 0 | 42.5 | 46.6 |
| *L. rhomboidalis* | *L. apiculatus* | 0 | 0 | 1.7 | 1.5 |
| *L. alveolatus* | *L. costaricensis* | 0 | 0 | 15.2 | 8.1 |
| *L. rhododendroides* | *L. schultzii* | 0 | 0 | 4.0 | 5.2 |
| *L. tenuifolius* | *L. ellipticus* | 0 | 0 | 17.1 | 20.1 |
| *L. oblongifolius* | *L. mutiscuanus* | 0 | 0 | 3.0 | 4.0 |
| *L. sp. nov. ANT* | *L. antioquensis* | 0 | 0 | 1.0 | 1.0 |
| *L. huertasii* | *L. julianii* | 0 | 0 | 7.0 | 7.0 |
| *L. eriophorus* | *L. chrysotrichus* | 0 | 0 | 7.0 | 9.0 |
| *L. colombianus* | *L. glutinosus* | 0.2 | 0.2 | 2.6 | 2.6 |
| *L. romeroi* | *L. saxatilis* | 1.0 | 0 | 3.0 | 1.5 |

further three pairs (21.4%) occur sympatrically and have evidence of ecological divergence in their leaf areas, while two cases (14.3%) present inconclusive evidence. When species not sampled in the phylogeny of *Vargas, Ortiz & Simpson (2017)* are considered (see Appendix S4) by pairing them to their most likely sampled sister species (row 2, Tables 1 and 2, Fig. 3), the contribution of allopatric speciation cases increases to 12 out of 15 (80%), while the number of cases for parapatric ecological divergence decreases to 1 (6.6%); the number of inconclusive cases remains about the same, 2 (13.3%) (Table 1). The aforementioned results are virtually the same to using grids of 0.1 and 0.05 decimal degrees to quantify range overlap (Table 3, Appendix S6). For example, range overlap calculated with a 0.1 grid produces the same results as our island framework in the analysis supplemented with missing sampling (Tables 2 and 3). The most fine-scale analysis, using a 0.05 grid decimal degrees, flips one of the pairs (*L. romeroi—L. saxatilis*) from sympatry to allopatry (this pair is likely to be parapatric with species occupying different vegetational belts), with this pair being the only conflicting result among the approaches.

We believe that the most reliable estimate for calculating the relative contribution of allopatric speciation *vs.* parapatric ecological divergence is the phylogeny supplemented with missing sampling on the island framework results (second row, Table 1; Table 2). These results agree with the 0.1 grid results, are corrected for missing sampling in the phylogeny, and are more conservative at estimating allopatric pairs (relative to using a 0.05
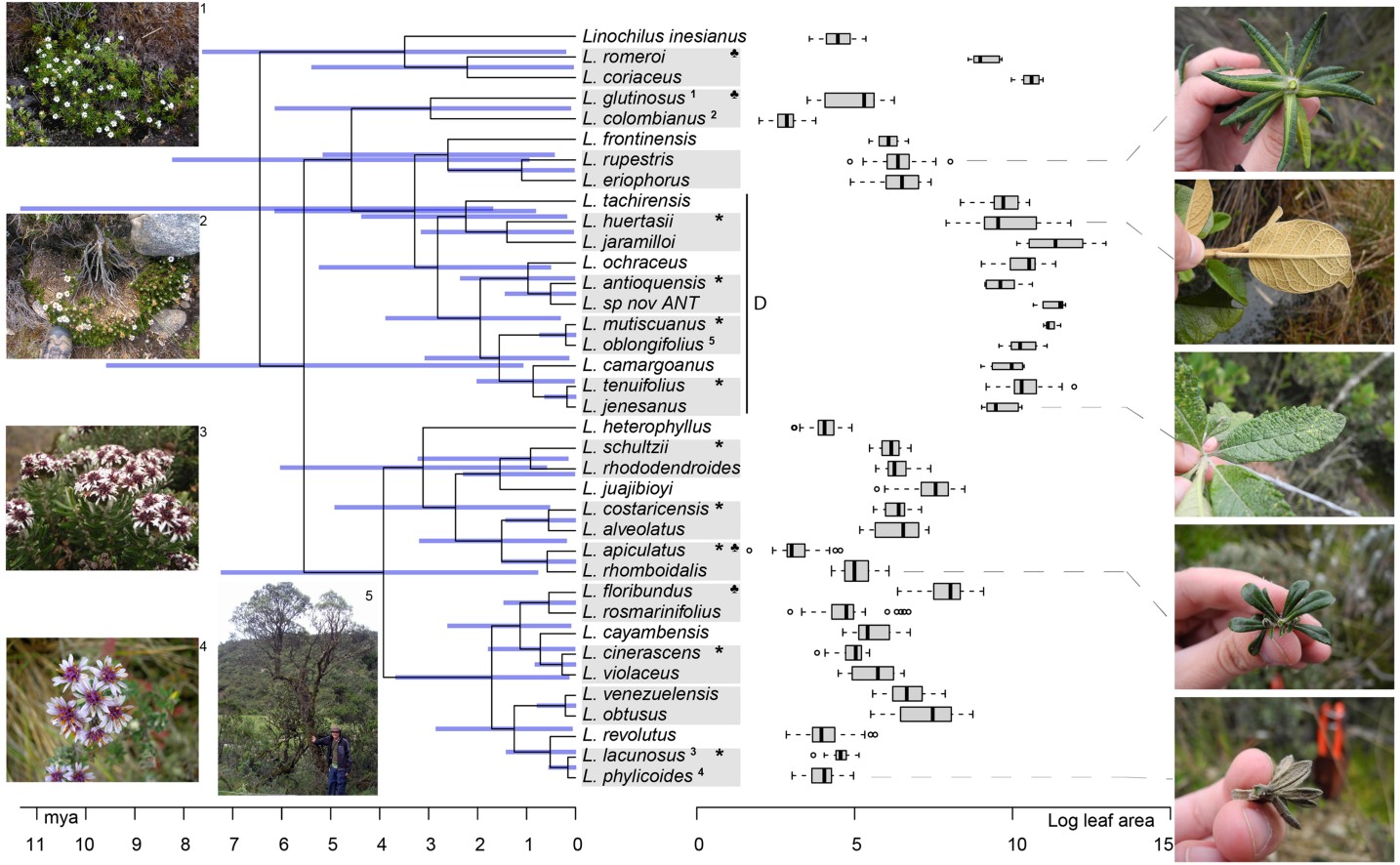

**Figure 4** *Linochilus* phylogeny (*Vargas, Ortiz & Simpson, 2017*) **with boxplots for leaf area.** Blue bars indicate confidence intervals for node ages. Photos on the left correspond to the superscript numeration on the tips of the phylogeny; photos 1 and 2 correspond to sympatric sister species with different leaf areas, while photos 3 and 4 correspond to allopatric sister species with similar leaf areas. Photos on the right correspond to the species indicated by the gray dashed line. The *Denticulata* clade is indicated by the letter D. Pairs are indicated by gray boxes, *: allopatric, ♣: different leaf areas.

degree grid). Therefore, we will focus our discussion on the species pairs derived from the phylogeny and substituted for missing sampling (Tables 2 and 3).

## Evaluating niche conservatism

Our results suggest that there is a strong signal of niche conservatism in the leaf area of *Linochilus*. Graphing the boxplots of leaf areas per species in front of the phylogeny reveals a general pattern in which closely related species tend to have similar leaf areas, likely occupying similar niches (Fig. 4). These observations are supported by the *Pagel's (1999)* lambda ($\lambda$) of 0.98 calculated with leaf area data on the phylogeny, $P = 2.8\mathrm{e}{-09}$ against the null hypothesis of $\lambda = 0$. This Pagel's lambda of almost one suggests that the evolution of the leaf area is highly dependent on the phylogeny, possessing a strong phylogenetic signal.

## Biogeographic analyses

The best scoring model in the biogeographic reconstruction for *Linochilus* was the BAYAREALIKE+J with an AICc of 226.78 followed by DEC+J with 232.62 (Table 4).

**Table 4 BioGeoBEARS results.** Comparison of the different biogeographic models for ancestral range inference evaluated by BioGeoBEARS in the phylogeny of *Linochilus* (*Vargas, Ortiz & Simpson, 2017*).

| Model | LnL | Num. params | AIC |
|---|---|---|---|
| DEC | −115.09205 | 2 | 234.184098 |
| DEC+J | −113.31082 | 3 | 232.621645 |
| DIVALIKE | −115.63941 | 2 | 235.278824 |
| DIVALIKE+J | −115.64036 | 3 | 237.280724 |
| BAYAREALIKE | −121.933 | 2 | 247.865999 |
| BAYAREALIKE+J | −110.39081 | 3 | 226.78162 |

The BAYAREALIKE+J reconstruction (Fig. 5) shows that the Colombian Eastern Cordillera played a major role in the diversification of *Linochilus*. This area (labeled E in Fig. 5), which contains the most species of the genus, was shown to be the ancestral range for most *Linochilus* clade ancestors (61%), dominating the backbone phylogeny. The ancestral range for the node representing the ancestor for all *Linochilus* species is inconclusive, with two areas sharing approximately two thirds of the relative probability: the Eastern Cordillera and the Northern Colombian Páramos (Sierra Nevada de Santa Marta + the Serranía del Perijá). The reconstruction shows that the *Denticulata* clade (clade D, Fig. 5), which colonized downslope to the upper limit of the cloud forest from the páramo, originated in the Eastern Colombian Cordillera. The biogeographic analysis also shows that approximately two thirds of the species sampled are restricted to a single páramo bioregion (27 spp., 71%), while the other third is found in two or more bioregions (11 spp. 29%). These biogeographic results evidence high endemism in the genus where most species are restricted to a few páramo islands, reinforcing the importance of geographic isolation in the diversification of the genus.

## DISCUSSION

In this study, we developed a framework to quantify the relative contributions of geographic isolation and parapatric ecological divergence in recent speciation events. We incorporated phylogenetics, geographic distributions, and a morpho-ecological trait. Our framework was applied to *Linochilus*, a genus of plants restricted to the páramo, which is the most species-rich tropical montane ecosystem. Because of the island-like distribution of the páramo and the within-páramo elevation gradient (Fig. 2), it has been suggested that allopatric speciation and parapatric ecological divergence are the main drivers of speciation in the high Andes (*van der Hammen & Cleef, 1986*; *Hughes & Atchison, 2015*). The application of our approach to *Linochilus* revealed that most recent speciation events are allopatric, with most sister species presenting allopatry and similar leaf areas (Tables 1–3).

### Allopatric *vs*. parapatric ecological speciation in the páramo

Our results suggest that most recent speciation events (12 events, 80%) in *Linochilus* are driven by allopatric speciation, while only a few are driven by parapatric ecological

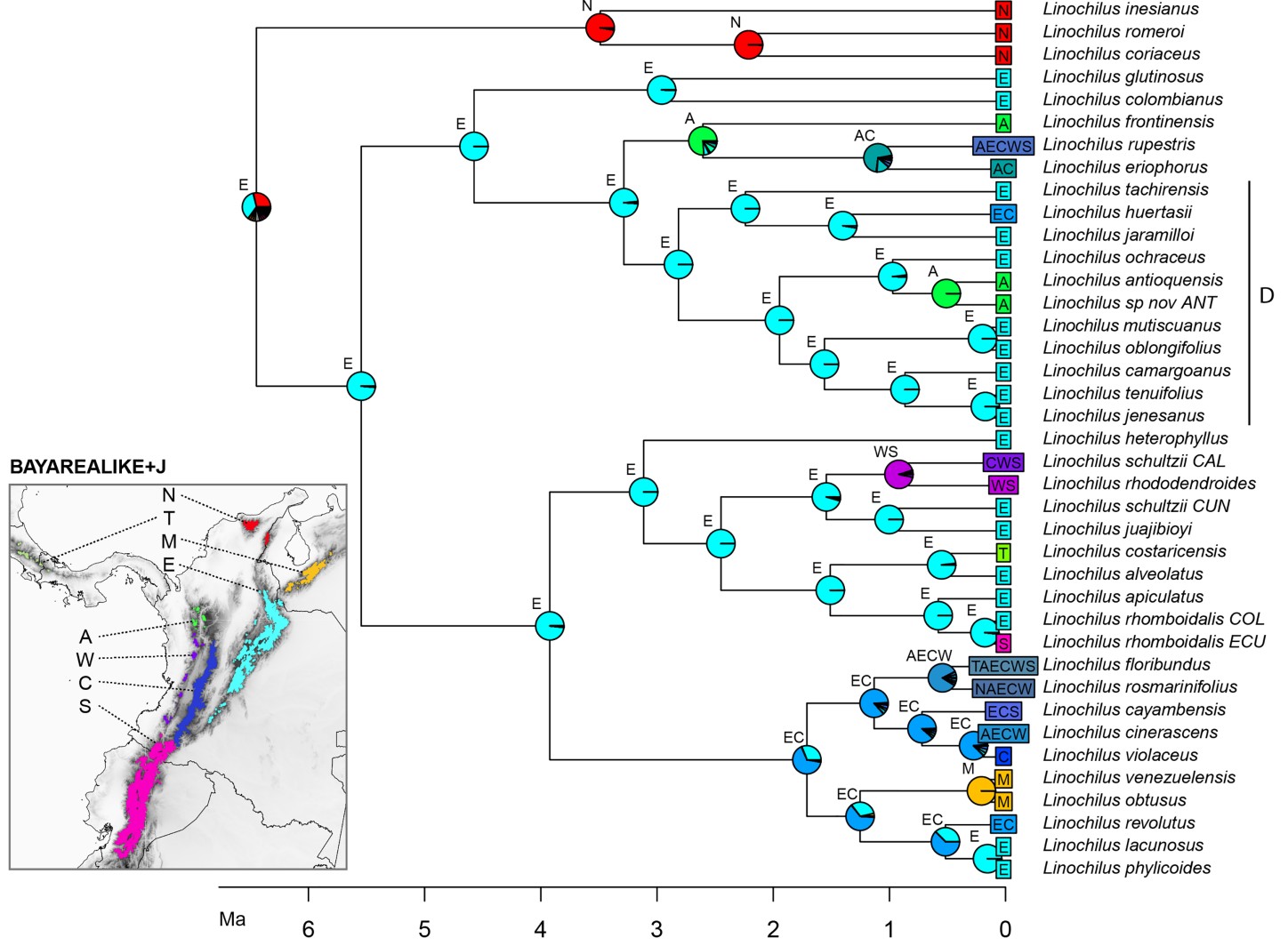

**Figure 5** BAYAREALIKE+J biogeographical ancestral reconstruction based on the *Linochilus* phylogeny of *Vargas, Ortiz & Simpson (2017)* with percent probabilities of the different ancestral areas as pie charts. Letters indicate biogeographic areas considered in the analysis. N, northern páramos; T, Talamanca; M, Mérida; E, Eastern Cordillera; A, Antioquia's páramos; W, Western Cordillera; C, Central Cordillera; S, southern páramos. Letters above each pie charts indicate the most probable area or area combination for that node.

divergence (1 event, 6.6%) (Table 2, Fig. 3). Out of the 12 allopatric pairs, only one shows a signal of subsequent ecological divergence. Inconclusive divergence in pairs (two events, 13.3%) was represented by a sympatric pair with similar leaf areas, and a sympatric pair with no leaf data. Independently of range overlap, only two pairs have significantly different leaf areas. This lack of ecological divergence suggests a strong signal of niche conservatism in *Linochilus*, as does a Pagel's λ of 0.98 (a value closer to one indicates strong phylogenetic signal in the evolution of leaf area). In the context of the páramo flora, our results are consistent with the hypothesis that páramos are island-like in promoting allopatric speciation (*Vuilleumier, 1971*; *Simpson, 1983*; *Simpson & Todzia, 1990*). Our results suggest that allopatric speciation alone could explain most of the recent and
rapid speciation found in other páramo genera where most species are restricted to one or few páramo islands: *Bartsia* (Orobanchaceae), *Espeletia* (Asteraceae), *Escallonia* (Escalloniaceae), *Hypericum* (Hypericaceae), *Jamesonia-Eriosorus* (Pteridaceae), *Lachemilla* (Rosaceae), and *Lupinus* (Fabaceae) (*Drummond et al., 2012*; *Zapata, 2013*; *Sánchez-Baracaldo & Thomas, 2014*; *Nürk et al., 2015*; *Uribe-Convers & Tank, 2015*; *Diazgranados & Barber, 2017*; *Contreras-Ortiz et al., 2018*; *Morales-Briones, Liston & Tank, 2018*; *Morales-Briones et al., 2018*).

Although our results suggests a minor role of parapatric ecological divergence (6.6%) (Table 1), our analysis focused on sister species that diverged recently and does not consider ancient divergent events. However, when we look at the distribution of leaf area on the phylogeny, we observe that taxa in clade D (*Linochilus* series *Denticulata*) have significantly larger leaves suited for dwelling in the upper zone of Andean forest (Fig. 3): Wilcoxon *P* < 2.2e−16 for both *L.* ser. *Denticulata vs.* its sister clade, and *L.* ser. *Denticulata vs. Linochilus*'s most species-rich clade (the clade originating with the most common ancestor of *L. heterophyllus* and *L. phylicoides*). The significantly larger leaves in *L.* ser. *Denticulata* suggests that an event of ecological shift, from microphyllous to macrophyllous leaves, took place ca. 3 mya near the ancestor of the *Denticulata* clade leading to the evolution of at least 20 species (32% out of all species in *Linochilus*). Based on the larger leaf area found in the *Denticulata* clade, which allows *Linochilus* species to be competitive in the upper zone of the cloud forest, we hypothesize that this downslope colonization event by the *Denticulata* clade is a rare case of ecological divergence.

Regardless of the reason for ecological divergence (after allopatric speciation or during parapatric ecological divergence), ecological divergence may boost allopatric speciation by allowing a lineage to colonize successfully a new niche and speciate in it by subsequent allopatric speciation in an island-like system. In the specific case of *Linochilus*, the evolution of larger leaves, which happened in one single event ca. 3 mya, allowed a lineage to colonize a lower vegetational belt and then speciate. A similar pattern is found in Andean *Senecio* (Asteraceae), where monophyletic forest and páramo clades have been documented to speciate in parallel (*Dušková et al., 2017*). In the context of adaptive radiations, we propose that ecological speciation happens fewer times when compared with allopatric speciation, but its effect could be significant at geological time scales; in other words, while allopatric speciation can explain most of the recent speciation, ecological divergence explains the origin of a single (or few) key adaptation(s) facilitating the colonization of new niches. *Espeletia*, whose crown origin is estimated at 2.5 mya, shows a peak in morphological differentiation relatively deep in its phylogeny at 1.5 mya (*Pouchon et al., 2018*) in addition to have a strong signal of allopatric speciation (*Diazgranados & Barber, 2017*). Other potential plant speciation boosters are genome duplication events (*Morales-Briones, Liston & Tank, 2018*) and pollination shifts (*Lagomarsino et al., 2016*).

## Alternative hypotheses and further considerations

Our framework is unable to distinguish between vicariant and dispersal speciation. Testing which kind of allopatric speciation occurs is challenging in island systems—typically

dispersal speciation predicts that the founder species will occur in a significantly smaller area than the source lineage (*Coyne & Orr, 2004*; *Anacker & Strauss, 2014*; *Grossenbacher, Veloz & Sexton, 2014*; *Skeels & Cardillo, 2019*), but geographically restricted patches (*e.g.*, páramos) can confound testing for significant differences in the distributions. Additionally, páramo islands have shifted their elevational distribution due to Pleistocene climate fluctuations, connecting páramo islands during glaciation periods and disconnecting them during interglacial periods (*Vuilleumier, 1971*; *Simpson, 1974*, *1975*; *Flantua et al., 2019*). This "flickering connectivity" could be a major driver of geographic isolation in páramo plants, especially for taxa with low seed dispersal ability (*i.e.*, *Espeletia* complex with ca. 100 spp.). In *Linochilus*, which has a fruit that easily disperses with wind, we find that 10 sisters (66.6%) present range asymmetry in which the widespread-sister range is >3 times larger than the small-ranged sister, suggesting dispersal speciation as a lead player in the diversification of *Linochilus*; we advise taking this result with caution for the aforementioned reasons. Finally, the best fit of the BAYAREALIKE+J model to *Linochilus* supports the idea that founder speciation events (dispersal speciation), modeled by the J parameter, are key for the speciation of *Linochilus*. While founder events make sense biologically considering the wind dispersal strategy in *Linochilus*, the validity of including the J parameter to biogeographic models has been challenged (*Ree & Sanmartín, 2018*, but see *Matzke, 2022*). Further studies at a phylogeographic scales could potentially shed light on the relative importance of vicariance *vs.* dispersal in the páramo.

Our sister-species framework assumes that speciation is a bifurcating process in which every speciation event produces two reciprocal monophyletic species. In the context of vicariant speciation in the páramo, a glacial-interglacial event could result in the fragmentation of one previously continually distributed population, into more than two daughter proto-species; the complex topography of the Colombian Eastern Cordillera provides a probable location for this process to occur (Fig. 2, Figs. 7–10 in *van der Hammen & Cleef, 1986*). Hypothetically, parapatric and dispersal speciation can also violate a bifurcating speciation model because a widely distributed population could be the source for multiple independent parapatric and or dispersal speciation events (*e.g.*, upslope colonization and adaptation in different mountains, multiple dispersal events, *Dexter et al., 2017*).

## Spatiotemporal history of *Linochilus* in the context of the páramo

Our biogeographic reconstruction of *Linochilus* shows that the genus originated in the Northern Andes 6.46 Ma, predating the estimated origin of the páramo 2–4 mya (*van der Hammen & Cleef, 1986*; *Gregory-Wodzicki, 2000*). Despite the fact that *Linochilus*'s inferred age 95% confidence interval of [1.71–11.37] can accommodate such incongruence, other primarily páramo genera like *Arcytophyllum* (Rubiaceae), *Brunfelsia* (Solanaceae), *Jamesonia+Eriosorus* (Pteridaceae), *Lysipomia* (Campanulaceae), and *Valeriana* (Caprifoliaceae) also show ages older than 4 mya (*Luebert & Weigend, 2014*).
An explanation for this early origin could be that ancestors of these páramo lineages inhabited the summits of middle elevation mountains (<2,000 m) extant at that time. Mid-elevation tropical mountains often have ridges that experience high winds and have

well-drained soils, making them physiologically dry. These mid-elevation patches pose similar physiological challenges to those of the páramo, as reported for contemporary *campos de altitude* and *campus rupestres* in Brazil (*Safford, 1999*; *Alves et al., 2014*). It is possible that middle elevation mountaintops provided an early habitat for *Linochilus* ancestors before higher elevations were available at 4 mya. A second alternative is that páramos were available before 2–4 mya as suggested by *Ehlers & Poulsen (2009)*. A third scenario is that *Linochilus* originated in the Sierra Nevada de Santa Marta (SNSM), a mountain range located in northern Colombia not considered part of the Andes Mountain Range. SNSM's paleoelevation remains largely unknown (*Villagómez et al., 2011*) and it is possible that this mountain had páramo prior to other mountains in north-western South America. Our BioGeoBEARS analysis provides some evidence for a SNSM *Linochilus* origin, as the analysis indicates the SNSM as the second most likely area of distribution for the most recent common ancestor of the genus (bioregion N, Fig. 5).

Our biogeographic analysis suggests that the Eastern Cordillera of Colombia played a major role in the diversification of the genus given the many extant and ancestral species whose distributional range include this area. The Eastern Cordillera contains the most páramo land area with discrete patches (*Londoño, Cleef & Madriñán, 2014*) making it ideal for autochthonous allopatric speciation. Our reconstruction also indicates that the Eastern Cordillera was the source for the colonization of three other mountain ranges: the Colombian Central Cordillera, the Colombian Western Cordillera, and the Talamanca Cordillera in Central America.

Achenes of *Linochilus* are small and a have a pappus that allows for long-distance dispersal of their seeds by wind (*Cuatrecasas, 1969*). Collectively, *Linochilus* is found on almost every páramo island, with the exception of the southernmost páramos, south of the Girón-Paute valley in Ecuador (*Jørgensen, Ulloa-Ulloa & Madsen, 1995*). Specific examples of long-distance dispersal in *Linochilus* are shown by the two species reported at the westernmost páramos in Costa Rica. *L. costaricensis*, endemic to Costa Rica, is probably a direct descendant of *L. alveolatus*, which inhabits the Colombian Eastern Cordillera. The second Costa Rican species, *L. floribundus*, is widespread, also reported for Colombia and Ecuador (*Vargas, 2011*, *2018*). The aforementioned examples of long-distance dispersal build on previous results about the potential dominant role of dispersal speciation in the genus and in other high montane taxa with seeds or fruits adapted to wind dispersal.

## CONCLUSIONS

Our comparative framework that incorporates phylogenetics, geographical distributions, and morpho-ecological characters unveiled a high signal of allopatric speciation, supporting it as the process driving most of the recent speciation events in *Linochilus*. The island-like distribution of páramo habitats is likely a primary factor of autochthonous allopatric speciation *via* geographic isolation, explaining the particularly high accumulation of plant species in the páramo (*Simpson & Todzia, 1990*) and their high speciation rates (*Madriñán, Cortés & Richardson, 2013*). Despite the comparatively small role of parapatric ecological speciation identified in recent *Linochilus* sister taxa, we

propose that ecological divergence has a role that is infrequent but potentially powerful in island-like systems. When ecological divergence does occur, it allows a lineage to colonize a new niche and then speciate by means of allopatric speciation among islands. Ecological divergence events that *boost* diversification are thus expected to be detectable at deeper geological time scales (>1 Ma). We conclude that geographic isolation and parapatric ecological divergence are positively synergistic processes in the history of the diversification of the páramo flora, contributing significantly to the global latitudinal species gradient.

## ACKNOWLEDGEMENTS

We thank Stefani Brandt, Julia Harencar, Shelley Sianta, Kathleen Kay, and Dena Grossenbacher for helpful comments and discussion during the writing of this manuscript. We also thank Melanie Tieje, Fabian Michelangeli, and an anonymous reviewer for their helpful comments and input. Santiago Madriñán thanks the Facultad de Ciencias, Universidad de los Andes, for general support. This study would not have been possible without the scientific collections housed at Universidad de los Andes Herbarium (ANDES), Herbario Nacional Colombiano (COL), the Field Museum (F), the Billie Turner Plant Resources Center (TEX), and the United States National Herbarium at the Smithsonian Institution (US).

### Funding

Financial support was provided by The University of Texas at Austin (Plant Biology Program Awards, the C. L. Lundell Chair of Systematic Botany, The Linda Escobar Award), the Garden Club of America (2012 Award in Tropical Botany), and the Smithsonian Institution (Cuatrecasas Award, 2006). The funders had no role in study design, data collection and analysis, decision to publish, or preparation of the manuscript.

### Grant Disclosures

The following grant information was disclosed by the authors:
The University of Texas at Austin (Plant Biology Program Awards, the C. L. Lundell Chair of Systematic Botany, The Linda Escobar Award).
The Garden Club of America (2012 Award in Tropical Botany).
The Smithsonian Institution (Cuatrecasas Award, 2006).

### Competing Interests

The authors declare that they have no competing interests.

### Author Contributions

- Oscar M. Vargas conceived and designed the experiments, performed the experiments, analyzed the data, prepared figures and/or tables, authored or reviewed drafts of the article, and approved the final draft.

- Santiago Madriñán conceived and designed the experiments, authored or reviewed drafts of the article, and approved the final draft.
- Beryl Simpson conceived and designed the experiments, authored or reviewed drafts of the article, and approved the final draft.

## Data Availability

The data is available at Bitbucket: https://bitbucket.org/oscarvargash/linochilus.

## Supplemental Information

Supplemental information for this article can be found online at http://dx.doi.org/10.7717/peerj.15479#supplemental-information.

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
