# Peer review of "Allopatric speciation is more prevalent than parapatric ecological divergence in a recent high-Andean diversification (Linochilus: Asteraceae)"

_PeerJ, doi:10.7717/peerj.15479_

## Round 0.1 · original submission · Minor Revisions

I sent the manuscript out for review to determine whether the referees would identify the merits of the study that would justify publication for PeerJ. After a careful reading of the manuscript and the reviewer reports, I would like to express my appreciation to the authors for their outstanding work. However, there are some points that must be clarified before a final decision is made. I hope that you will find all advice helpful when revising the manuscript.

·

Basic reporting

Basic reporting
This study adds new evidence for the modes of plant speciation in a spatially heterogeneous biodiversity hotspot. Derived from phylogenetic, distribution and trait data of sister species, the study uses biogeographic models that suggest that allopatric speciation is prevalent for this Asteraceae clade.
The manuscript cites existing literature and provides enough background information to place it in the research field. The structure of the manuscript is well organized, and I only have one suggestion regarding the structure, as well as some more smaller comments on material and methods description (see detailed comments below).

Experimental design

Experimental design
The study is a relevant contribution to the field and fits the scope of the journal. Technical standard is adequate. Some methods would profit from a more detailed description (comments on this below). I would also recommend particularly checking the statistical analysis for the leaf area comparison on which the niche analysis is based on (detailed comment below as well).

Validity of the findings

Validity of the findings
Reproducibility: The authors provide raw data for occurrences, not for the leaf measurements though, those would need to be added. Code is not provided, could be added as well.

Additional comments

General comments
I commend the authors for providing the raw occurrence data, I still suggest to provide the occurrence data points used in the analysis as machine-readable csv/txt or similar. That would be a big help for anyone wanting to work with the data once it is published. Adding R-code would also help reproducibility.

Methods structure:
I really like the entire sister species model for investigating speciation mechanisms, however it is also a tricky one that (among other factors) depends on the quality of the underlying phylogeny and the completeness to make sure sister species are actually sister species. This issue was transparently addressed using alternative parings from literature, however the phylogeny based approach was introduced more prominently throughout the text, yet only holds the smaller number of species and the authors themselves deem the substituted data more reliable. It almost looks a bit like the authors throw the initial analysis based on their phylogeny overboard by adding the substituted dataset, so I think for clarity it would make more sense to a) state these two approaches earlier in the methods, and then make the substituted dataset the focus and the phylogeny-only sister pair analysis as a sensitivity analysis maybe?

Age and speciation as temporally distinct event:
Incorporating age of sister species pairs seems like a low hanging fruit here with ages being provided in Table 2, since speciation events are events in flux, the timing might tell about how far range filling / ecological divergence has progressed – especially relevant for an area that has very high div rates as the authors mention in the introduction. Are sister species pairs with ecological divergence older?
Related to this, more of a personal comment than a request for additional analysis: Since the authors notice the” strong geographic isolation with little ecological divergence” as stated in lines 265f, have the authors thought about a more continuous approach to allo-/sympatry scoring for the analysis? Using the amount of overlap in range as a numeric variable instead of grouping species into one of two categories might give, especially in combination with species age, additional information about the in-between stages of speciation. I am aware that using categories makes analysis much more straight forward, however I cannot help but think that we are missing information in the continuity of all ecological processes… Does ecological divergence strength correlate with divergence time?

Leaf area measures:
How do leaf area and plant specimen age compare, I would think younger plants have smaller leafs? I would also assume the herbarium material will comprise mostly adult specimens, but it would be great to include a short note on this potential confounding variable in the respective part of the Material section (i.e. line 132ff)


Detailed comments
Line 96: The here cited reference is about the Diplostephium phylogeny, only later (line 119ff) it is mentioned that the nomenclature has changed and part of Diplostephium is now Linochilus. Maybe that sentence would fit better closer to the citation of the phylogeny.

Line 133: I noticed two things in Appendix S3; 1) there are two species with even more than 30 leafs measurements, 2) the variation in number of individuals and total leaf number is quite big, that is understandable and not a problem since the Wilcoxon test (at least rank sum) is robust towards sample size differences – but I wonder if the variation in measured leafs could be presented somewhere, e.g. in Figure 4 as boxplot width corresponding to the sample number (varwidth=TRUE) in the boxplots on the right. If that looks too messy, mean + SD in the text would be great too.

Line 144: I do not think those are non-independent samples as in “paired”, at least not between the two compared species. The two compared groups(=species) are independent, but the data within the species are nested: up to 6 subgroups(=individual) per species with x number of leaf measures per subgroup, so those measures per individual are non-independent. I am not sure what this does to the stats without digging into literature, but I am sure that the individual here is a grouping variable which needs to be accounted for. A quick search showed an R package “nestedRanksTest”, maybe that could be helpful, or suitable ANOVA if the log-transformation results in sufficient enough normalization.

Line 147ff: From the initial description, I didn’t immediately understand that presence absence from a mountaintop and grid cell range overlap are two different approaches and was wondering how contrasting cases were handled (e.g., species do occur on the same mountaintop but have no overlap?). Phrase so that it becomes clear that its 2 (3) different ways to score distribution.

Line 250ff: I am fairly familiar with a range of biogeographic models but have never used BioGeoBEARS myself and I feel like these models are kinda prone to appearing as black boxes. Since the other method parts were described in great detail I would appreciate one or two sentences about how the model works, something like “the model takes x & y as input and performs action z to calculate the ancestral area along the chronogram.”


Minor things
I noticed páramo is spelled with and without the accent throughout the manuscript, unless intentional, stick to one.

Line 30ff: Percentages do not add up to 100%, so currently I think that just distracts from the main message, maybe skip numbers in the abstract all together?
Line 42: remove dash from “hot-spots”
Line 83: 2-4 Ma somehow lacks the “ago” part, in paleo/geology one would write “mya”, maybe check journal guideline for that (or simply use “million years ago”)
Line 99: double “this”
Line 105: remove template placeholder text ;)
Line 216: one verb too many for extinction rates
Line 233f: For convenience, could the authors add here how the biogeographic areas were defined according to Londoño et al. (2014), skips the extra step of having to look it up.
Line 261: I find it usually very helpful to provide a short data summary at the end of methods, to know how many pairs we are dealing with here. Something like “We identified the modes of speciation for 14 (15 for substituted data) sister species pairs using grid cell and presence-absence based approaches distribution data across 8 mountaintops”.




##### Figures
Fig 1: I noticed that the caption of figure 1 speaks of both leaf shape and leaf size, stick to one phrase.
Fig 3: The double cross symbol in the figure lacks description in the caption.


#####
An interesting and sound study!

Best,

Melanie Tietje, Aarhus University

Reviewer 2 ·

Basic reporting

See Additional Comments

Experimental design

See Additional Comments

Validity of the findings

See Additional Comments

Additional comments

Review of MS for PEERJ – Manuscript ID 81187 – Vargas et al., Allopatric speciation is more prevalent than parapatric ecological divergence in a recent high-Andean diversification (Asteraceae: Linochilus).

The 31st of January 2023

This manuscript aims at disentangling the respective role of allopatric and parapatric speciation in the diversification of Linochilus, a genus distributed in the Paramo, a biodiversity “hotspot”. Therefore, the authors proposed a careful argument based on a test developed to discriminate between both speciation mode.

I appreciated this MS and the rigor of the argument. Indeed, the hypotheses and assumption behind the analyses are clearly stated. The limits of the developed test are identified and explicitly mentioned. The discussion is balanced and do not “oversell” the results. The writing is clear and concise.

I only have a few minor comments (see below) and I recommend accepting this manuscript.

I would only suggest detailing why, in the method section, you used a log-transformed leaf areas (implying the use of Wilcoxon signed-rank test), rather than just the leave area (likely normally distributed and a t-test).

Minor comments/suggestions
L38: “218 years” instead of “217 years”
L44: biodiversity hotspot and topographic complexity: are the last paper on that topic really dating back to Jenkins et al. 2013?
L68: maybe it should be precise here that “sister species” have a common ancestor, as it might not be clear for all readers.
L97: I would write the name in full letter rather as an acronym. (technically it is abbreviation and each letter should be followed by a “.”.
L105: remove “Add your introduction here”
L110: How is the fact that the genus is “almost entirely restricted to the paramo” in favour of your hypothesis?
L162: I suggest to replace “… in the following fashion” by “We interpreted the results as follow (Fig.1):”
L195: replace the ugly abrreviation “A.K.A.” by “also known as”
L257-259: indeed, but have you tried a constrain model?
L353-355: please consider rewriting this sentence as the use of the conjunctions is unclear (Although our results …., our analysis…. “here results and analysis are the same”).
L380-381. You cannot, just add polyploidization and pollination shift as potential diversification booster, without making a link with your study. What about the chromosome numbers of Linochilus? Do we know them? Are they constant (which would be help full for your argumentation) or not? What about pollinator, do we know them, etc…
L430-432: this statement is unclear. Why would the SNSM be the second (and not e.g. the first) most likely area of origin? It only is a sister clade! There is many other plausible scenari explaining the current distribution.
L434: what do you mean by “ancestral species”? do you mean “common ancestor of some living species”? please clarify.
L465-466: remove “Add your conclusions here”.


I hope those comments may be useful to the authors.

·

Basic reporting

Vargas and collaborators present a very interesting study to determine whether allopatric or sympatric speciation are responsible for diversification dynamics in the high Andes. To this effect they use a previously published phylogeny of the genus Linochilus a group in the sunflower family mostly restricted to the Paramos or high elevation cloud forest, coupling it with species distribution and comparison of these distributions across putative sister taxa. The study also includes a biogeographic analysis of the genus and leaf size as a proxy for niche differentiation.
The authors do a great job in presenting the problem, and two sections are particularly welcomed: 1-The clear explanation of how the results were interpreted (lines 162-181 and Fig. 1); 2-clearly stating the assumptions involved in the study (lines 190-220)( but see below).

Experimental design

There are four areas of the study that need either revision or better justification:
1-Given the low medium density sampling of the phylogeny I was a bit concerned about identifying sister species pairs. The authors attempt to get around this by adding putative species pairs based on morphology. To justify this better perhaps an explanation of how the phylogeny matches morphology (provided in Vargas et al. 2017 but not here) and state whether a similar exercise with the species actually sampled in the phylogeny would yield a similar result.
2-Interpretation of the biogeographical analyses (particularly DEC models) can be heavily influenced by the optimization at the base of the focal clade. Why not incorporate the information available on the sister clade to Linochilus?
3-The entire section of range size asymmetry is not clearly justified and perhaps not even necessary. As it is, it detracts from the study. I see two issues here: A-Current distribution range may not reflect distribution ranges during critical times when speciation/diversification occurred given the cyclical nature climate in the Andes, specially during the last 2 MY as species have moved up and down the Andean elevational gradient (as already cited in their introduction); B-Even if current range size is a reflection of the range size at the time of speciation of species pairs, the relative size may be governed by factors not responsible for the speciation event (for example, the given area of a paramo is simply much larger or smaller that the paramo occupied by its sister species), and what is more important is whether the distributions overlap or not (as this study sets out to determine).
4-For species that the ranges only overlap on their edges and are considered sympatric, a better justification of why this could not be a case of allopatric speciation with secondary meeting zone is perhaps needed.

Validity of the findings

Nothing to add (see below)

Additional comments

In addition to the comments above, there are additional comments on the manuscript and below (numbers refer to the lines in the PDF):
48: Vicariant or peripatric…Perhaps “dispersal” fits better here than peripatric.
49-55: This could be shortened. For the purposes of speciation dynamics (and this paper) what matters is the pattern of distribution of the species (allopatric or sympatric) and not really how that was achieved (vicariance or dispersal).
74-75 need more justification why it provides an ideal setting
97 provide reference of the monograph? (Vargas, in prep.)
102-105: The question about allopatry vs sympatry applies equally in paramos as in cloud forest. I don’t understand what the authors mean here. Perhaps rewrite?
108-112 I would suggest to exchange the positions of the 1st and 2nd sentences. First explain the distribution of the genus then add that a phylogeny is available.
112 after introducing the phylogeny please proved details about it (type and number of markers, number of species, % of known species, etc)
120: on what base were segregated the 2 genera? (Vargas et al. 2017 perhaps?, but make it clear)
140 what is Denticulata?
139-144: add explanation as to why the Wilcoxon test can’t be applied on the sister species assumed based on morphology and not sampled in the phylogeny.
154-155 I don’t understand why you use overlap = intersection/range of smaller instead of overlap = intersection/range of both. Ok got it, it is explained just after.
172 Or sympatric speciation no?
191 split this sentence
256 the choice of using the +J models should be better justified. As cited (Ree and Sanmartín 2018) its use is debated. Need also to precise how the models were evaluated and how the best model was chosen (AIC?).
263-264 I don’t understand the beginning of the sentence.
284 Evaluating niche conservatism: this paragraph includes sentences that would suit better in the discussion than in the results
323-329 is results, not discussion
330 this paragraph could be in results
338-341numbers in parentheses are confusing, I suggest “(12 events, 80%)” instead of “(12, 80%)”
347 ref needed
348 could explain, instead of can explain
365 rare case of ecological divergence
367 remove italic for boost
371 Provide the family for Senecio and Espeletia (and all other mentioned genera)
380-381 Ok but here you do not infer diversification per se
397-400 You cannot say that without discussing the bias towards more founder events when incorporating the J parameter, as suggested by Ree RH, Sanmartín I (2018) Conceptual and statistical problems with the DEC+J model of founder-event speciation and its comparison with DEC via model selection. Journal of Biogeography 45: 741–749. https://doi.org/10.1111/jbi.13173

Other considerations:
What about reconstructing ancestral state for leaf area? (micro vs. macrophyllous leaves) what about habit shift as well?
Is there anything known about ploidy levels, phenology or pollination biology that would explain species isolation for those sympatric cases?

---

## Round 0.2 · accepted · Accept

I would like to express my appreciation to the authors for their careful review and I am happy to accept this manuscript in its current form.